# The Role of Autophagy as a Trigger of Post-Translational Modifications of Proteins and Extracellular Vesicles in the Pathogenesis of Rheumatoid Arthritis

**DOI:** 10.3390/ijms241612764

**Published:** 2023-08-14

**Authors:** Gloria Riitano, Serena Recalchi, Antonella Capozzi, Valeria Manganelli, Roberta Misasi, Tina Garofalo, Maurizio Sorice, Agostina Longo

**Affiliations:** Department of Experimental Medicine, “Sapienza” University of Rome, 00161 Rome, Italy; gloria.riitano@uniroma1.it (G.R.); serena.recalchi@uniroma1.it (S.R.); antonella.capozzi@uniroma1.it (A.C.); valeria.manganelli@uniroma1.it (V.M.); roberta.misasi@uniroma1.it (R.M.); tina.garofalo@uniroma1.it (T.G.); agostina.longo@uniroma1.it (A.L.)

**Keywords:** rheumatoid arthritis, post-translational modifications, extracellular vesicles, exosomes, autophagy

## Abstract

Rheumatoid arthritis (RA) is a chronic systemic autoimmune disease, characterized by persistent joint inflammation, leading to cartilage and bone destruction. Autoantibody production is directed to post-translational modified (PTM) proteins, i.e., citrullinated or carbamylated. Autophagy may be the common feature in several types of stress (smoking, joint injury, and infections) and may be involved in post-translational modifications (PTMs) in proteins and the generation of citrullinated and carbamylated peptides recognized by the immune system in RA patients, with a consequent breakage of tolerance. Interestingly, autophagy actively provides information to neighboring cells via a process called secretory autophagy. Secretory autophagy combines the autophagy machinery with the secretion of cellular content via extracellular vesicles (EVs). A role for exosomes in RA pathogenesis has been recently demonstrated. Exosomes are involved in intercellular communications, and upregulated proteins and RNAs may contribute to the development of inflammatory arthritis and the progression of RA. In RA, most of the exosomes are produced by leukocytes and synoviocytes, which are loaded with PTM proteins, mainly citrullinated proteins, inflammatory molecules, and enzymes that are implicated in RA pathogenesis. Microvesicles derived from cell plasma membrane may also be loaded with PTM proteins, playing a role in the immunopathogenesis of RA. An analysis of changes in EV profiles, including PTM proteins, could be a useful tool for the prevention of inflammation in RA patients and help in the discovery of personalized medicine.

## 1. Introduction

Rheumatoid arthritis (RA) is a chronic systemic autoimmune disease, characterized by persistent inflammation of the diarthrodial joints, which can ultimately direct to cartilage and bone destruction, resulting in pain, disability, and a reduction in life expectancy. RA is a multifactorial disease determined by both genetic and environmental factors and impacts approximately 0.5% to 1% of adults worldwide; women and the elderly population show a higher incidence of RA [1,2,3,4,5]. Since rapid development results in irreversible joint damage, optimal management of RA is mandatory soon after disease onset [6].

RA progresses from a preclinical state to early synovitis and, finally, to destructive disease [7]. The preclinical phase of RA, without clinical evidence, precedes by many years the clinical onset of the disease. In this phase, a loss of tolerance, immune activation, and epigenetic remodeling lead to the early generation of autoantibodies, which can bind post-translationally modified self-proteins, particularly via citrullination. This can be facilitated by environmental risk factors, such as cigarette smoking and/or mucosal microbiota disturbance [8,9]. A transition event that involves a "second hit", though poorly understood, allows for the development of clinically and imaging-detected synovitis (early RA) [7,8]. This phase is characterized by the activation of T cells, B cells, and macrophages, which infiltrate the synovial membrane, activating synoviocytes and synovial inflammation with the amplification of an autoantibody, cytokines (IL-6, IL-1, and TNF-α), and chemokine production. These events cause joint destruction and an established RA [8,9,10,11,12]. RA patients’ neutrophils are more prone to spontaneously undergo NET formation, which may induce immunogenicity [9,13]. Moreover, the receptor activator of nuclear factor kappa-Β ligand (RANKL) produced by synovial fibroblasts, as well as certain T- and B-cell populations, induce the differentiation of monocytes into bone-resorbing osteoclasts. The Janus kinase (JAK)/STAT signaling pathway amplifies the immune response and the activation of nuclear factor kappa-light-chain-enhancer of activated B cells (NF-kB) in these inflammatory cells [9].

However, synovial histological and transcriptional analyses have shown marked heterogeneity among patients with established RA; therefore, approaches based on systems biology might also help in patient stratification according to the different pathogenic pathways for treatment personalization [14]. Therefore, drugs that interfere with the previous pathogenic mechanisms have been shown to be useful in the treatment of RA patients [9,15,16,17,18,19,20,21,22,23,24,25,26,27,28] (Table 1).

Serological diagnostic testing, in particular the presence of autoantibodies, is of growing importance in the early detection and differentiation of RA [29,30,31]. At present, IgM Rheumatoid Factor (RF) and anti-citrullinated protein antibodies (ACPAs) are used in routine serodiagnosis [32]. Both RF and ACPAs have been included in the ACR-EULAR 2010 classification criteria for RA and are employed as biomarkers for diagnostics [33]. Their presence defines seropositive patients [34]. Usually, seropositive RA is associated with an increment in joint damage and radiographic progression, while seronegative RA patients have higher inflammation parameters at presentation [35,36,37]. ACPAs are detected in nearly 70% of RA patients [38]. Their diagnostic sensitivity in early arthritis is 67%, but they are much more specific than RF, by around 85–95% [9]. ACPAs are present before the onset of RA symptoms, and their identification in patients has the highest predictive value for development of RA [39]. ACPAs are associated with more severe joint disruption and the development of earlier and abundant erosions in comparison with patients without ACPAs [2,40]. ACPAs seem to be associated with cardiovascular disease and mortality [41,42]. ACPA-positive RA patients respond better to treatment in an early-RA phase of the disease but achieve drug-free remission less frequently [2]. Therefore, ACPA-positive patients need an aggressive initial approach to prevent radiographic progression [43]. Moreover, it has been discovered that post-translational modifications of ACPAs seem to contribute to the pathogenesis of RA, such as the extensive glycosylation of the IgG ACPA V domain, or a decrease in Fc galactosylation and an increase in the Fc fucosylation of serum ACPA IgG1, which predisposes one to RA development [37,44,45]. The recent identification of novel anti-carbamylated protein antibodies (anti-CarP) suggested the hypothesis that the diagnostic gap including seronegative patients could be closed [46,47]. Anti-CarP antibodies are identified in up to 45% of RA patients. The sensitivity of these antibodies is 18–26% and 27–46% before and after RA diagnosis, respectively, and their specificity is around 90% in RA patients [37,48]. Anti-CarP IgG antibodies seem to be associated with a more severe radiological progression in ACPA-negative RA [49]. Recently, the identification of anti-CarP antibodies was also associated with a higher disease activity and more disability over time in RA patients [28]. The identification of anti-CarP antibodies in patients with arthralgia predicts the development of RA regardless of the onset of ACPAs [49,50,51,52]. Therefore, anti-CarP antibodies might be a useful biomarker to identify in ACPA-negative patients who have a diagnosis of early-RA patients and require early and aggressive clinical intervention [53]. Despite the identification of all these autoantibodies, there is still a need to improve the diagnosis of RA. Since the absence, deficiency, or excess of post-translational modifications may evolve in the generation of autoantigens that can lead to autoimmune responses, with a loss of tolerance to the self, further studies in this direction could help to discover novel effective biomarkers for early diagnosis and prognosis in order to achieve better disease management of RA patients.

## 2. Post-Translational Modifications of Proteins in RA Patients

Post-translational modifications (PTMs) encompass a group of reactions that modify the structure and extend the functions of proteins. Indeed, PTMs are chemical changes that can be mediated by enzymes or can result from non-enzymatic additions that recognize specific target sequences in specific proteins [54,55]. These modifications involve a physiological process in which the covalent addition of functional groups to protein occurs to maintain their structure, function, and stability, but they might also be related to protein ageing [56].

Post-translationally modified proteins play a role in the pathogenesis of several autoimmune diseases, such as RA, systemic lupus erythematosus (SLE), and antiphospholipid syndrome (APS), as they can generate various autoantigens. Furthermore, we demonstrated that post-translationally modified proteins are already present on the surface of circulating extracellular vesicles (EVs) [57].

In RA, the key role of PTMs is evidenced by the observation reported above that ACPAs are considered as the main biomarkers for diagonosing RA. However, besides citrullination, several other PTMs are also involved in the generation of autoantibodies, such as carbamylation, acetylation, and oxidation [56,57,58,59] (Table 2).


**
CITRULLINATION:
**


Protein citrullination is an irreversible PTM and refers to the process of conversion of peptidyl-arginine to peptidyl-citrulline. Citrulline is also a metabolite of the urea cycle, and citrullination is catalyzed by the peptidyl arginine deiminase (PAD) enzyme family. The loss of a positive charge caused by citrullination produces electrostatic and conformational changes of the modified protein, affecting its function by altering binding sites, protein–protein interaction, and susceptibility to degradation [60], and, as mentioned above, the formation of a citrullinated protein also suggests the possibility of generating new epitopes that could act as a new autoantigen that escapes self-immune tolerance [61].

Increased PAD activity, and thus increased protein citrullination, is strongly linked to the progression of RA [62]. Indeed, citrullinated peptides are present in RA, and anti-citrullinated protein antibodies are the serological markers for the diagnosis of RA [36]. The principal citrullinated proteins in RA patients are alpha-enolase-1, vimentin, and type II collagen [35].


**
CARBAMYLATION:
**


Homocitrullination or carbamylation is a non-enzymatic reaction between isocyanic acid and free amino groups of proteins, and involves the conversion of lysine to homocitrulline. Among other factors, inflammation, oxidative stress, a high level of urea, and tobacco smoke seem to induce protein carbamylation [63].

In patients with RA, it has been demonstrated that the presence of antibodies directed against carbamylated proteins may be useful in predicting higher disease activity and may be related to inflammatory biomarkers [64]. As citrullination, the principal carbamylated proteins in RA patients are alpha-enolase-1, vimentin, and type II collagen [57].


**
ACETYLATION:
**


Acetylation is a reversible enzymatic process (a balance between acetylases and deacetylases) in which acetyl group donors, such as acetyl-CoA and acetyl phosphate, are added covalently to free amines of lysine residues or to the N-terminus protein. Mechanically, negatively charged acetyl groups covalently add to specific lysine residues in proteins that can decrease their electrostatic affinity. Acetylation and deacetylation, coordinated by acetyl-transferase and deacetylase, are in a dynamic balance to maintain normal physiological and biochemical processes of cells. However, when the balance is broken, pathological processes will result [56].

Lysine acetylation has also been shown to play a key role in immune system regulation [64]. This modification occurs on different proteins. In particular the presence of acetylated-lysine vimentin has been demonstrated in RA patients [65]. Acetylation may also regulate autophagy, a physiological process contributing to the maintenance of cellular homeostasis by the degradation of unnecessary or dysfunctional components through a lysosome-dependent regulated mechanism.


**
OXIDATION:
**


Oxidative stress is defined by an imbalance between reactive oxygen species (ROS) production and impaired detoxification by antioxidant enzymatic and non-enzymatic systems. Pro-oxidation conditions can cause conformational changes in protein structures by promoting PTMs. When there is an overproduction of ROS and/or a deficiency of the antioxidant machinery, a biochemical imbalance occurs and causes tissue damage. The pathogenic role of oxidative stress and inflammation are also related to their ability to effect protein structural modifications [66,67].

Several groups have suggested a role for oxidative stress in the pathogenesis of RA and demonstrated increased oxidative enzyme activity, as well as decreased antioxidant levels in RA sera and synovial fluids. Indeed, studies of RA synovial fluid and tissue demonstrated oxidative damage to hyaluronic acid, lipid peroxidation products, oxidized low-density-lipid (LDL) proteins, and increased carbonyl groups reflective of oxidation damage to proteins [59]. The principal protein in RA modified by oxidants including •OH, HOCl, and ONOO− is type-II collagen [68].

As many proteins in the joint are long-lived, especially proteins expressed in cartilage, it is not surprising that some post-translationally modified proteins are found at higher levels in the joint compartment, which might also contribute to its vulnerability as a target for PTMs [53]. The principal proteins of inflamed joints and synovium in RA are type II collagen, fibrinogen, fibrin, vimentin, and alpha-enolase-1 [69]. Through a prolonged and/or enhanced exposure of post-translationally modified proteins, the synovial compartment could become susceptible to an immune response recognizing these modified proteins, thereby leading to an increased risk of a chronic inflammatory response [9,53]. Furthermore, we demonstrated that PTM proteins are already present on the surface of circulating EVs in patients with RA, indicative of a higher expression of citrullinated antigen, which has shown a significant correlation with disease activity [57].

**Table 2 ijms-24-12764-t002:** Post-translational modifications associated with rheumatoid arthritis and their self-antigens.

Modification	Principal Antigens in RA	Modified Aminoacid	Implication	Ref.
**Citrullination**	Vimentin, alpha-enolase-1 type II collagen	Arginine	Induces modifications in protein conformation. Exposure of neoepitopes to the immune system with increase in its immunogenicity.	[57,58,60,61,62]
**Carbamylation**	Vimentin, alpha-enolase-1 type II collagen	Lysine	Changes in structure of proteins inducing an immune response specific for a modified protein.	[57,63,64]
**Acetylation**	Vimentin	Lysine, N-terminus protein	Induces an immune response specific for a modified vimentin and exposes the cryptic epitope promoting antibody binding.	[56,64,65]
**Oxidation**	type II collagen	Cysteine, Lysine	Change in protein structure with an increase in its immunogenicity.	[59,68]

## 3. The Role of Autophagy on Post-Translational Modifications of Proteins in RA Patients

The autophagy process could be involved in post-translational changes of proteins and in the generation of citrullinated [62] and carbamylated [47] peptides recognized by the immune system in RA, with a consequent breakage of tolerance [40]. In this way, autophagy may represent a key processing event, creating a substrate for autoreactivity.

Autophagy is described as a regulated process inside almost every cell type activated against various stress conditions, such as starvation, protein aggregation, hypoxia, oxidative stress, and endoplasmic reticulum (ER) stress [70]. At the basal level, autophagy contributes to control biological process, the quality of proteins, and organelles, and eventually leads to a safe environment for cells [71]. Thus, damaged organelles, impaired and misfolded proteins, protein aggregates, and intracellular pathogens are encapsulated into autophagosomes and then fused with lysosomes for subsequent degradation [72]. Autophagy is a stress response that allows unicellular eukaryotic organisms to survive during harsh conditions, probably by regulating energy homeostasis and/or by protein quality control [71]. Alterations in autophagy machinery may be implicated in autoimmune diseases [73,74]. In particular, a significant difference in autophagic propensity between T lymphocytes from healthy donors and patients with SLE has been observed, demonstrating that lymphocytes from SLE patients are more resistant to autophagy induction [75]. Defective autophagy was also observed in the chondrocytes of patients with Kashin-Beck disease [76], in which ATG4C was identified as a susceptibility gene [77].

Moreover, RA synovium exhibits a highly increased ER stress-associated gene signature [78] and TNF-α further increases the expression of ER stress markers in fibroblast-like synoviocytes (FLSs) [79]. Kato et al. [80] identified a dual role for autophagy in the regulation of stress-induced cell death in RA FLSs. Interestingly, citrullination in the autophagosomes may increase the catabolism of the proteins, as charged residues of the proteins are eliminated. Thus, a key role for autophagy in the citrullination of peptides by antigen-presenting cells has been hypothesized [81,82,83]. We demonstrated in vitro a role for autophagy in the citrullination processes [84]. Indeed, autophagic cells showed PAD-4 activation, with consequent protein citrullination. Ex vivo, a significant association between levels of autophagy and anti-CCP antibodies was observed in naïve RA patients.

As reported above, a number of environmental conditions, including smoking and infections, are associated with RA [83]. Autophagy may be the common feature by which, in several types of stress (including smoking, joint injury, and infections), autophagy vesicles of antigen-presenting cells may drive a response to citrullinated peptides recognized by the immune system [83]. In addition, we demonstrated that autophagic cells show a significant increase in carbamylated proteins, and a significant correlation was found between autophagy and carbamylation levels in mononuclear cells of naïve RA patients [47]. These studies support the view that post-translational processing of proteins in autophagy may generate autoantigens recognized by the immune system in early-active RA (Figure 1).

## 4. The Interplay between Autophagy and Exosomes

Several studies have been undertaken to understand the crosstalk between endomembrane organelles and the molecular mechanisms involved in vesicular trafficking. Vesicular processes are highly dynamic and closely depending on subcellular compartments. Among them, the well-known vesicular processes are autophagy-related vesicles [84] and endosome-derived vesicles, i.e., exosomes [85]. The latter represent a class of nano-sized EVs, which derive from endosomal compartments and share several lines of linkages with endocytosis, lysosomal degradation, and autophagocytosis.

The capacity of EVs derived from the endosomal system to interact with the autophagic process has been extensively reported [86]. In this regard, biochemical studies support the evidence that autophagy shares with the molecular machinery of EVs, which include autophagy-related proteins and key proteins for EV biogenesis and secretion pathways [87,88]. In fact, unlike degradative autophagy, the autophagic machinery, including ATG factors, may lead to a form of unconventional secretion/expulsion of cytosolic proteins instead of their degradation; this mechanism appears to be of particular importance for protein secretion, immune surveillance, and cell signaling [89,90].

Thus, whether under physiological or pathological conditions, the crosstalk between exosome–autophagy networks ensure the cellular homeostasis via the lysosomal degradative pathway and/or the secretion of cargo into the extracellular space [91].

Exosomes, which are small EVs, have emerged as key players in the development and progression of RA-related joint inflammation. These unique EVs perform essential functions by facilitating the transportation of autoantigens and mediators between distant cells within affected joints [92].

Exosome biogenesis is a tightly regulated process. The molecular machinery includes four multiprotein complexes, known as the endosomal sorting complexes responsible for transport (ESCRT-0, -I, -II, and -III), in addition to auxiliary molecules. The cascade of interaction among ESCRT subunits and accessory molecules leads to the budding of vesicles into endosomes [93]. In mammalian cells, multivesicular body (MVB) generation is affected by autophagic machinery.

Recently, three different forms of autophagy have been extensively investigated: (i) macroautophagy refers to the formation of double membrane vesicles named autophagosomes, which enclose proteins and/or organelles, delivering them to the lysosome for degradation; (ii) microautophagy refers to the direct engulfment of cellular components to be degraded by lysosomes; (iii) chaperone-mediated autophagy (CMA) refers to the transport of target proteins to lysosomes in a lysosome-associated membrane protein-dependent manner. Macroautophagy is essential for the regulation of cellular function, organelle degradation, and adaptation to stress. The others are more directly involved in the fine regulation of cellular function.

To date, over 30 proteins encoded by specific Autophagy-related genes (Atg) are mandatory for macroautophagy, the most studied type of autophagy (hereafter referred to as autophagy). Among these, Autophagy-related-5 (ATG5), Autophagy-related-7 (ATG7), Autophagy-related-12 (ATG12), in association with Autophagy-related-16-like-1 (Atg16L1), participate in an enzymatic cascade that drives the nucleation, expansion, and closure of the phagophore in response to various stress stimuli. In particular, the macromolecular ATG12-ATG5-ATG16 complex, is responsible for the covalent modification of LC3-I (microtubule-associated protein 1 light chain 3, ATG8) with the amine part of phosphatidylethanolamine to form LC3-II, which is essential for autophagosome formation. Next, autophagosomes can fuse with lysosomes enabling their cargo to be degraded by acidic hydrolases. Alternatively, autophagosomes can also be fused with endosome-derived cell structures, such as MVBs. The key role of autophagy-related proteins, including ATG16L1 and ATG5, in exosome biogenesis in normal and pathological conditions has been well-determined [94]. For instance, ATG5 promotes the process leading to the fusion of MVBs with the plasma membrane in breast cancer cells; the inhibition of the ATG16L1 and ATG5-ATG12 complex markedly affects exosome biogenesis or their secretion, in addition to the sorting of LC3, a well-known autophagic marker, into exosomes [95]. Taken together, ATG5 and ATG16L1 protect MVBs from lysosomal degradation and direct them into the secretory pathway instead of the lysosomal pathway. Interestingly, the interaction of ATG12 with ATG3, which is responsible for LC3β conjugation, regulates exosome biogenesis through interaction with apoptosis-linked gene 2-interacting protein X (ALIX), a protein that cooperates with the ESCRT-III complex. Interestingly, the inhibition of ALIX decreases the autophagy flux, indicating a regulatory cross-link between exosome biogenesis and autophagy pathways [96].

In addition, ESCRT-independent machinery, including several lipids (i.e., ceramide), tetraspanins (CD9, CD63, and CD81), and other proteins, plays a pivotal role in the biogenesis of MVBs and exosome sorting [97].

As reported above, the exosome cargo may also contain molecules sorted from vesicles generated during the autophagic process named autophagosomes [98], though other intracellular vesicular systems, such as Golgi apparatus/vesicles, in the endocytosis pathway are not excluded. Crosslink between exosome biogenesis and autophagy pathways that engages the vesicular system has been supported by the biogenesis of hybrid vesicles inside cells referred to as amphisomes [99]. These vesicles are generated through the fusion of MVBs with autophagosomes, which finally combine with lysosomes for the hydrolysis and degradation of cargo; alternatively, they fuse with the plasma membrane for releasing intraluminal vesicles (ILVs) in extracellular space [100]. Based on this evidence, it can be assumed that MVBs represent transient structures, where cellular conditions affect their fate for degradation versus secretion. In addition, it is interesting to note that the fate of autophagosomes can also shift from a conventional degradation pathway to a secretory one depending on cellular conditions.

Interestingly, recent evidence suggests that autophagy actively provides information to neighboring cells via a process called secretory autophagy. Furthermore, the autophagy and lysosomal/exosomal secretory pathways have been demonstrated to serve as a canal to degrade and expel damaged molecules out of the cytoplasm to maintain homeostasis and preserve cells against stress conditions [91]. Secretory autophagy combines the autophagy machinery with the secretion of cellular content via EVs (Figure 2).

## 5. Role of Exosomes and Microvesicles in RA Pathogenesis

Numerous papers have demonstrated a role for exosomes in RA pathogenesis [101,102]. Exosomes are involved in intercellular communications, and some reports [103,104] have found upregulated proteins and RNAs inside them that contribute to the progression of RA. Some studies have shown that the total number of exosomes in both plasma and synovial fluid is increased in RA patients compared to healthy individuals [102].

Exosomes are present in the synovial fluid of inflamed joints, which originate from FLSs and cells infiltrated in the synovial joint, including platelets, granulocytes, monocytes, neutrophils, and T and B cells [105,106]. Exosomes from RA FLSs were shown to promote their abnormal proliferation and synovial hyperplasia [107]. These exosomes contain membrane-bound forms of TNF-α that in turn promote the activation of NF-kB and the induction of membrane-type matrix metalloproteinase (MMP)-1 in RA FLSs [108]. RA FLSs release transforming growth factor beta (TGF-β), enhancing RA FLS proliferation and angiogenesis. In addition, the increase of the 24- and the 17–18-kDa Toll-like receptor (TLR) 3 fragments has been observed in serum exosomes of RA patients, which may reflect the hyperactive state of RA [109]. The TLR3 signal activates NF-κB and Interferon Regulatory Factor (IRF) 3 transcription factors, which lead to the secretion of type I interferons and proinflammatory cytokines, such as IL-6 and IL-8 [110]. Interestingly, four microRNA (miRNAs), i.e., miR-155-5p, miR-146a-5p, miR-323a-5p, and mir-1307-3p, were upregulated upon TNF-α stimulation in the exosomes derived from FLSs, and different studies have shown the role of these miRNAs in the pathogenesis of RA [111]. Furthermore, post-translationally modified proteins, mainly citrullinated proteins, known as autoantigens in RA, were detected in exosomes purified from the synovial fluids of RA patients [112]. These citrullinated proteins, such as the Spα receptor, the fibrin α-chain fragment, the fibrin β-chain, the fibrinogen β-chain precursor, the fibrinogen D fragment, and vimentin enhance the production of pro-inflammatory cytokines and initiate pro-inflammatory responses characterized by Th1 and Th17 proliferations [112,113]. According to many reports [92,112,113,114], circulating exosomes have shown an ability to present citrullinated peptides to the effector cells in the form of an MHC-peptide complex. In contrast, some exosomes like those derived from mesenchymal stem cells (MSCs) can decrease joint destruction, suppressing FLS proliferation and promoting cartilage regeneration, such as exosomes containing miR-150-5p [115]. Furthermore, T cell-derived exosomes containing miR-204-5p could contribute to the inhibition of FLS proliferation [116]. Exosomes from different sources can affect RA progression by inducing the proliferation of CD4+ T cells and their differentiation towards Th17 cells, a pro-inflammatory cell population, in RA. For example, miR-424 in exosomes derived from RA FLSs significantly induced Th17 differentiation and inhibited Treg cell differentiation under hypoxic conditions [117]. On the other hand, miR-146a and miR-155 in MSC exosomes suppress T- and B-cell immune responses and increase Treg in vitro [118]. Moreover, exosomal miR-155 and miR-146a can be used for the early diagnosis of RA. Additionally, miRNA17 was upregulated in exosomes purified from RA patients’ plasma, which can suppress Treg induction by inhibiting the expression of transforming growth factor-beta receptor II (TGFBR II) in RA patients [118,119].

Another important factor in RA pathogenesis is bone resorption, and some exosomes may promote this process. Previous studies have shown that the levels of RANKL in exosomes isolated from the synovial fluid of patients with RA were significantly higher than those of patients with several other types of arthritis and induced higher numbers of osteoclasts involved in bone destruction [120]. In addition, exosomes from FLSs contain increased levels of miR-221-3p and mir-92a that can induce bone destruction in RA patients [121]. Correspondingly, the expression level of Hotair, a kind of long non-coding (Lnc) RNA leading to the migration of active macrophages, was greater in the exosome from RA patients. Hotair induces the release of MMP-2 and MMP-13 by osteoclasts and synoviocytes. Furthermore, Hotair is quite stable and easily detected in blood and urine and could be used as a diagnostic marker for RA [122]. In theory, the selective elimination of these exosomes would be beneficial to arthritis therapy.

Macrophages are relevant in the pathogenesis of RA and activated macrophages found in RA synovia are an early hallmark of RA. The effect of exosomes on macrophages in RA is still relatively limited. The secretion of miR-let-7b-containing exosomes promotes the differentiation of M1-type pro-inflammatory macrophages in RA joint inflammation [107]. Serum-derived exosomes miR-6089 and miR-548a-3p can regulate macrophage proliferation and differentiation [107,123].

These observations show the presence of disease-contributing exosomes, which could be useful inflammation markers of arthritis diseases. Furthermore, it has been shown that a subset of RA patients contains IgM RF associated with EVs, among which are exosomes, which can be used to distinguish between active and inactive RA [124].

Several proteomic studies have been performed to identify the exosomal components and their potential functions in the development of inflammatory arthritis. In RA, most of the exosomes are produced by the leukocytes and synoviocytes, and they are loaded with inflammatory molecules and enzymes that might be implicated in RA pathogenesis and the inflammatory process; therefore, they could be used as markers for RA subsets.

In addition to exosomes, MVs derived from cell plasma membrane may also play a role in the immunopathogenesis of RA [125]. The mechanisms by which MVs originate from plasma membrane are not fully known; external stimuli such as calcium ionophore, collagen, and epinephrine, as well as stress and mechanical factors, lead to the release of MVs. An influx of Ca^2+^ as an exogenous stimulus and the release of calcium from the endoplasmic reticulum leads to the activation of calpain, which plays an important role in the formation of MVs by participating in the reorganization of the cytoskeleton, which in turn participates in the shedding of MVs. The formation of MVs and their subsequent release is very often linked to the translocation of phosphatidylserine to the outer membrane of the cells. Moreover, the release of MVs can occur in specialized microdomains of the plasma membrane, i.e., lipid rafts, areas enriched in cholesterol and specialized in signal transduction, as they are rich in proteins involved in cell activation. MVs can transfer a series of information from one cell to another; they themselves can carry different molecules depending on the cell type from which they originate, influencing the functions of the cells they meet [126,127].

The presence of MVs has been demonstrated in biological fluids in both health and disease conditions. They can influence different functions in different diseases, contributing to their pathogenic mechanisms. Following cell activation or cell death (apoptosis, necroptosis, pyroptosis, and NETosis), large numbers of MVs are released into the blood, such as in the case of autoimmune diseases, including SLE and APS. Autoantigens generated during apoptosis are redistributed into the membrane surface of MVs or apoptotic bodies [57].

In rheumatic disorders, MVs isolated from synovial fluid have been shown to negatively impact osteoarthritis (OA) and RA disease progression [113]. Transmission electron microscopy observations demonstrated the occurrence of large multilamellar synovial MVs that are altered in synovial fluid from OA and patients with RA. There is also a difference in the biochemical properties of the synovial fluid of patients with OA and RA joints as compared to human samples collected from healthy volunteers [128]. In particular, the protein amount present in the synovial fluid is a greater than two-fold increase in OA samples and a greater than 2.5-fold increase in RA samples, as compared to healthy volunteers [128]. In addition, it has been shown that the number of MVs from the plasma of RA patients is significantly higher than in healthy donors [57]. In these MVs from RA patients, we identified three principal proteins modified: vimentin, alpha-enolase1, and type II collagen [57].

A large presence of autoantigens typical of RA, including carbamylated or citrullinated proteins, are contained in the MVs released by patients, together with proinflammatory cytokines, which contribute to endothelial activation, such as adhesion molecules and chemokines. Moreover, it has been observed that MVs released by patients with RA promote an M1 macrophage profile, with a consequent amplification of the pro-inflammatory clinical picture [107]. Citrullinated neoepitopes have been described as key triggers of ACPA synthesis in patients with RA.

Platelets are the main source of MVs in blood, and their presence improves communication within the immune system; furthermore, platelets are involved in the crosstalk between the immune system and the coagulation system. In RA, platelets and platelet-derived MVs have been detected in both blood and synovial fluid samples [129].

Expression of both the enzyme PAD-4 and citrullinated proteins was demonstrated for the first time in human platelets and platelet-derived products (PDPs). In addition, ACPA-mediated platelet activation has been observed in RA patients. Both platelet aggregates and microparticles released as a consequence of platelet activation have been observed in joints in patients with RA [129]. However, the mechanistic events leading to platelet activation in RA have not yet been well characterized. It was hypothesized that platelet-citrullinated proteins and PDPs may play a prominent role in stimulating platelet activation in RA. Indeed, many citrullinated proteins found in platelets and PDPs can be recognized by ACPAs, and these autoantibodies can stimulate platelet activation, leading to the release of inflammatory active molecules and citrullinated autoantigens that can sustain inflammatory responses in RA joints.

## 6. Conclusions

Recent evidence shows the role of autophagy in the immunopathogenesis of RA, indicating its role in both MV biogenesis and protein post-translational modification triggering (Figure 2).

In the last few years, EVs, including exosomes and MVs, have drawn attention due to their multiple roles in health and disease conditions. Proteomic studies have prompted the identification of MV/exosome components and their potential functions in the development of inflammatory arthritis. In RA, most of the EVs are produced by leukocytes and synoviocytes and are loaded with proinflammatory molecules that might play a role in the inflammatory process and in RA pathogenesis. Thus, an analysis of changes in EV profiles, including the post-translational modification of proteins, could be a useful tool for the prevention of inflammation in RA patients and help in the discovery of personalized medicine.

## Figures and Tables

**Figure 1 ijms-24-12764-f001:**
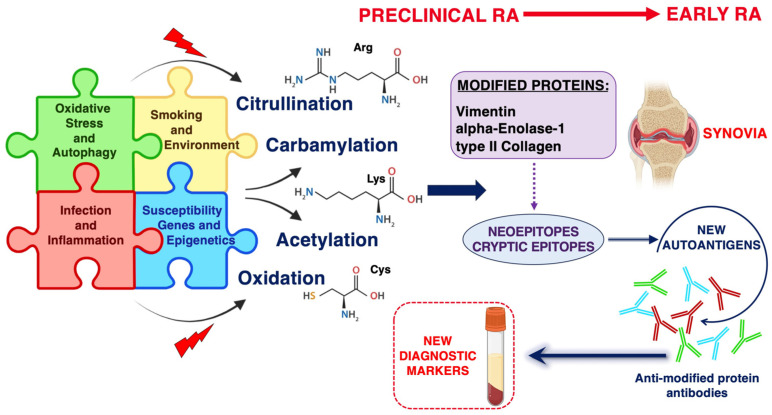
Post-translational modifications of proteins involved in RA immunopathogenesis.

**Figure 2 ijms-24-12764-f002:**
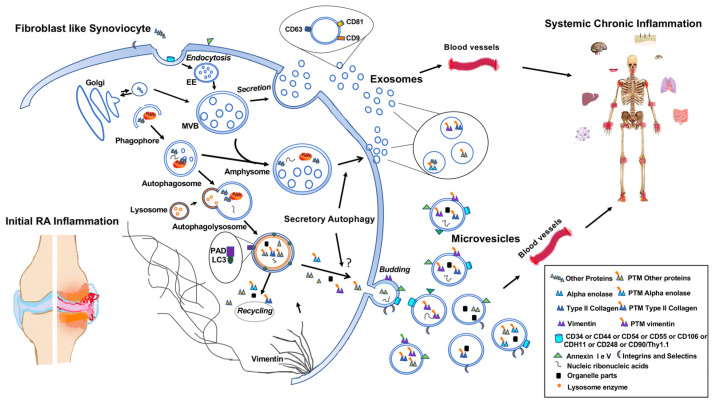
Schematic drawing depicting the interplay between autophagy and the generation of EVs and their role in the inflammatory process and in RA pathogenesis.

**Table 1 ijms-24-12764-t001:** Main drugs used for the treatment of rheumatoid arthritis.

Generality	Therapy	Mechanism of Action	Ref.
**Common Initial Treatments**	NSAIDs	Inhibition of inflammation.	[16]
	Corticosteroids: Short term Glucocorticoids (GCs)	Upregulation of anti-inflammatory and downregulation of pro-inflammatory genes.	[15,17]
	Conventional synthetic DMARDs (i.e., Methotrexate (MTX), Hydroxychloroquine (HCQ), Sulfasalazine, Leflunomide, Azathioprine)	Inhibition of immune cell proliferation (all DMARDs); stabilization of macrophage lysosomes (HCQ); inhibition of different pathways, including adenosine metabolism (MTX, Sulfasalazine, and Leflunomide).	[15,18,19]
**Biological DMARDs**	TNF inhibitor (i.e., Infliximab, Adalimumab, Etanercept, Certolizumab, Golimumab)	Inhibition of cells activation, preventing matrix degradation and the production of proinflammatory molecules.	[15,18,19]
	IL-inhibitors (i.e, IL-6R: i.e., Tocilizumab; Sarilumab)	Inhibition of IL-mediated signaling and its pro-inflammatory effects.	[15,19]
	B-cell depleting agent (CD20: i.e., Rituximab)	Depletion of B cells, inhibiting antigen presentation and autoantibody production.	[15,20]
	T-cell-costimolatory blocking agents (i.e., Abatacept)	Binding with CD80 and CD86 and blocking T-cell costimulation, inhibiting naive T-cell activation.	[15,21]
	CXCL chemokine inhibitors (future)	Inhibition of chemokines and chemokine receptors.	[22]
**Targeted Synthetic DMARDs**	JAK-inhibitors (i.e.,Tofacitinib, Baricitinib, Upadacitinib, Filgotinib)	Interruption of cytokine networks through blockade of the JAK–STAT pathway, inhibiting FLS activation, leukocyte maturation, and autoantibody production.	[15,23]
**Cell Therapy**	Mesenchymal stem cell (MSC) therapy (future)	Modulation of the immune response via cell-to-cell communication and MSC-secreted cytokines.	[24]
**Epigenetic** **Therapy**	DNA-methyltransferase (DNMT) and Histone-deacetylase (HDAC) inhibitors (future)	Modification of epigenetic, restoring abnormalities in RA.	[25]
**Targeting** **Autophagy**	Autophagy regulators (i.e., Rapamycin, Chloroquine (CQ), Hydroxychloroquine (HCQ) 3-Methyladenine (3-MA) (future)	Activation of autophagy by inhibiting mTOR (Rapamycin); autophagy suppression, reducing the activity of T cells and apoptosis resistance (CQ and HCQ); inhibition of autophagy at an early stage of autophagosome development by blocking P13K signaling (3-MA).	[15,26,27,28]

## Data Availability

Not applicable.

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
