# Peer review of "The Role of Autophagy as a Trigger of Post-Translational Modifications of Proteins and Extracellular Vesicles in the Pathogenesis of Rheumatoid Arthritis"

_ijms, 2023, doi:10.3390/ijms241612764_

Round 1

Reviewer 1 Report

Please enrich the introduction part by adding more information regarding the RA and its pathogenesis: more updated references , more details about treatments etc.

Author Response

Please enrich the introduction part by adding more information regarding the RA and its pathogenesis: more updated references, more details about treatments etc.

We improved the introduction section, by adding more information regarding pathogenesis and treatment. Eight new references were added (new Refs. 7-14).

We also added a new Table 1 summarizing the main drugs used in the treatment of RA and their relationship with the pathogenic mechanisms (new Refs. 15-28).

Reviewer 2 Report

In the paper entitled "Role of Post-Translational Modifications of Proteins and Extracellular Vesicles in the Pathogenesis of Rheumatoid Arthritis," the authors performed an exhaustive review of the implications of post-translational modifications of proteins in the development of a very common autoimmune disease, Rheumatoid Arthritis.

The manuscript is well-written and easy to read. However, there are some parts of the paper that need to be corrected to create a publishable version of this work. More specifically, the paper has both minor and major issues.

Minor issues:

-In section 2, “Post-translational modifications of proteins in RA patients,” the concept of autophagy is discussed from line 154, but it is not briefly defined until line 213. Please, kindly provide a concise introduction of the autophagy after its initial mention to clarify its meaning.

-In section 3, "The role of autophagy on post-translational modifications of proteins in RA patients," the role of autophagy in the post-translational modification of proteins in the disease is explained. However, the different types of autophagic pathways, called microautophagy, macroautophagy, and chaperone-mediated autophagy (CMA), are not mentioned. As a result, the concept of autophagy is not clearly defined. Please introduce and explain these autophagic pathways.

-The role of autophagy is mentioned other human diseases such as SLE (Systemic Lupus Erythematosus), but not in other arthritis-like diseases such as Kashin-Beck disease. Please mention and discuss this association. DOI: 10.1016/j.joca.2016.09.019 ¸ DOI: 10.1016/j.joca.2014.08.010

-From lines 284-299, the functions of some genes, such as ATG5-ATG16L1, ATG12, ATG3, are discussed in specific contexts related to exosome formation. However, these genes are not mentioned earlier, nor is their main function in the canonical pathway (macroautophagy) of autophagosome formation explained. I recommend adding an additional figure that explicitly mentions or explains the key genes involved in the autophagic pathway for autophagy formation. Alternatively, these issues could be incorporated into the text, or a figure with a detailed explanation could be included in the legend.

Major issues:

-In my opinion to conduct a comprehensive review of the pathogenesis of rheumatoid arthritis, it is necessary to include a table outlining the main drugs used for its treatment. This table would assist readers in gaining a better understanding of the current state of the art concerning the disease.

-Figure 2 is too small, blurry, and difficult to see. Please improve the quality of the figure and also adjust the font size.

Author Response

In the paper entitled "Role of Post-Translational Modifications of Proteins and Extracellular Vesicles in the Pathogenesis of Rheumatoid Arthritis," the authors performed an exhaustive review of the implications of post-translational modifications of proteins in the development of a very common autoimmune disease, Rheumatoid Arthritis.

The manuscript is well-written and easy to read. However, there are some parts of the paper that need to be corrected to create a publishable version of this work. More specifically, the paper has both minor and major issues.

Minor issues:

-In section 2, “Post-translational modifications of proteins in RA patients,” the concept of autophagy is discussed from line 154, but it is not briefly defined until line 213. Please, kindly provide a concise introduction of the autophagy after its initial mention to clarify its meaning.

We added a concise introduction of the autophagy after its initial mention.

-In section 3, "The role of autophagy on post-translational modifications of proteins in RA patients," the role of autophagy in the post-translational modification of proteins in the disease is explained. However, the different types of autophagic pathways, called microautophagy, macroautophagy, and chaperone-mediated autophagy (CMA), are not mentioned. As a result, the concept of autophagy is not clearly defined. Please introduce and explain these autophagic pathways.

The concept of autophagy and the different types of autophagic pathways have been improved (see line 267 and following).

-The role of autophagy is mentioned other human diseases such as SLE (Systemic Lupus Erythematosus), but not in other arthritis-like diseases such as Kashin-Beck disease. Please mention and discuss this association. DOI: 10.1016/j.joca.2016.09.019 ¸ DOI: 10.1016/j.joca.2014.08.010

We reported this association in Section 3, lines 211-213, and added two new references (n. 76-77). 

-From lines 284-299, the functions of some genes, such as ATG5-ATG16L1, ATG12, ATG3, are discussed in specific contexts related to exosome formation. However, these genes are not mentioned earlier, nor is their main function in the canonical pathway (macroautophagy) of autophagosome formation explained. I recommend adding an additional figure that explicitly mentions or explains the key genes involved in the autophagic pathway for autophagy formation. Alternatively, these issues could be incorporated into the text, or a figure with a detailed explanation could be included in the legend.

We clarified in the text the key genes involved (section 4, lines 276-287).

Major issues:

-In my opinion to conduct a comprehensive review of the pathogenesis of rheumatoid arthritis, it is necessary to include a table outlining the main drugs used for its treatment. This table would assist readers in gaining a better understanding of the current state of the art concerning the disease.

We agree with the Reviewer and added a new Table 1 summarizing the main drugs used in the treatment of RA.

-Figure 2 is too small, blurry, and difficult to see. Please improve the quality of the figure and also adjust the font size.

We improved the quality of Figure 2 and increased the font size.

Reviewer 3 Report

The topic of review is interesting and relevant in RA. The review lacks focus and comprise of so many irrelevant details which are already published in previous review articles. For instance, a review published by Kwon et al 2021 already comprises of information regarding post translational modification of protein in RA. Therefore, the review may focus on how autophagy promotes generation of post-translational modification and extracellular vesicle formation which are associated with RA.

Their diagnostic sensitivity in early arthritis is 57%, but they are much more specific than RF, around 96% (8). What is 8 here? Line 61

Line: 101 repetition of a word.

The introduction needs to shortened and more concise.

Line 111  translationally

Line 130 citrullinated

Line 160 pro-oxidation

Line 182 significative of

Point 2 seems irrelevant

The review may start from point 3 with oxidative stress as one of sub headings.

Line 250-283 can be shorten.

Line 284-311 I think is not related with the review topic.

Line 333-345 is not needed

Line 445-448 please specify the control subjects involved in the study 

line 415-421 is not needed. 

There are so many mistakes in the sentences. The English language needs extensive editing.

Author Response

The topic of review is interesting and relevant in RA. The review lacks focus and comprise of so many irrelevant details which are already published in previous review articles. For instance, a review published by Kwon et al 2021 already comprises of information regarding post translational modification of protein in RA. Therefore, the review may focus on how autophagy promotes generation of post-translational modification and extracellular vesicle formation which are associated with RA.

We agree that previous articles summarized information regarding post translational modification of protein in RA. In this concern, we also added the Reference of Kwon (new Ref. 9). However, the main novelty of our review article is the role of autophagy as a trigger of post-translational modification and extracellular vesicle formation which are associated with RA. Thus, we modified the title, accordingly.

Their diagnostic sensitivity in early arthritis is 57%, but they are much more specific than RF, around 96% (8). What is 8 here? Line 61

Sorry, it was the number of the Reference (old Ref. 8).

Line: 101 repetition of a word.

We corrected the text.

The introduction needs to shortened and more concise.

Reviewers 1 and 2 required more information and a new Table in the introduction section.

Line 111  translationally

We corrected the text.

Line 130 citrullinated

We corrected the text.

Line 160 pro-oxidation

We corrected the text.

Line 182 significative of

We corrected the text.

Point 2 seems irrelevant

The review may start from point 3 with oxidative stress as one of sub headings.

We think that a complete description can be useful.

Line 250-283 can be shorten.

We shortened this section.

Line 284-311 I think is not related with the review topic.

We shortened this section. However, the key role of ATG16L1 and ATG5 in exosome biogenesis in normal and pathological conditions has been well-determined. Thus, according to the indication of Reviewer 2, we clarified in the text the key genes involved (lines 276-287).

Line 333-345 is not needed

We deleted these lines.

Line 445-448 please specify the control subjects involved in the study 

We specified the control subjects involved in the study (healthy subjects). 

line 415-421 is not needed. 

We deleted these lines.

Round 2

Reviewer 2 Report

Considering the modifications made by the authors, I believe that the manuscript is now ready to be published.

Author Response

We thank the reviewer for his/her positive comment.

Reviewer 3 Report

Line 445-448 please specify the control subjects involved in the study. Why will a healthy individual require synovial fluid aspiration ? the subjects in control group had some other issues such as reactive arthritis or injury? Please specify.

Author Response

Line 445-448 please specify the control subjects involved in the study. Why will a healthy individual require synovial fluid aspiration ? the subjects in control group had some other issues such as reactive arthritis or injury? Please specify.

In this study (Reference n. 128) 7 healthy human samples were collected from healthy volunteers. In the paper it is indicated that samples were purchased by Leebio (Maryland Heights, MO, USA).

We modified the text accordingly (highlighted in green).